# A Proposed Method of Converting Gait Speed and TUG Test in Older Subjects

**DOI:** 10.3390/ijerph191912145

**Published:** 2022-09-25

**Authors:** Joanna Kostka, Natalia Sosowska, Agnieszka Guligowska, Tomasz Kostka

**Affiliations:** 1Department of Gerontology, Medical University of Lodz, Milionowa 14, 93-113 Lodz, Poland; 2Department of Geriatrics, Healthy Ageing Research Centre, Medical University of Lodz, Hallera 1, 90-647 Lodz, Poland

**Keywords:** sarcopenia, frailty, walking speed, Timed Up and Go, functional status

## Abstract

Sarcopenia is one of the most important health problems in advanced age. In 2019, the European Working Group of Sarcopenia in Older People (EWGSOP) updated the operational diagnostic criteria for identification of people with sarcopenia (EWGSOP2). Among the two proposed low performance measures of sarcopenia are gait speed and the Timed Up and Go (TUG) test. Usage of any of those tools requires recalculation for the second one for eventual comparisons. The simple linear regression has been used for such comparisons in several previous studies, but the appropriateness of such an approach has not been verified. The aim of this study is to find the most appropriate model describing the relationship between these two popular measures of physical function. The study was performed in 450 consecutive outpatients of the Geriatric Clinic of the Medical University of Lodz, Poland, aged 70 to 92 years who volunteered to participate in the study. The TUG test and gait speed at 4 m to assess physical function were used. Different alternative models were compared to obtain the highest R-squared values. A Reciprocal-Y model (R-squared = 71.9%) showed the highest performance, followed by a Logarithmic-Y square root-X model (R-squared = 69.3%) and a Reciprocal-Y square root-X model (R-squared = 69.1%). The R-squared for the linear model was 49.5%. For the selected reciprocal model, the correlation coefficient was 0.85 and the equation of the fitted model was: Gait speed (m/s) = 1/(−0.0160767 + 0.101386 × TUG). In conclusion, in independent community-dwelling older adults, the relationship between gait speed and the TUG test in older subjects is nonlinear. The proposed reciprocal model may be useful for recalculations of gait speed or TUG in future studies.

## 1. Introduction

Walking is a dynamic activity that requires the use of an appropriate muscle function. Age-related sarcopenia, associated with a gradual decrease in muscle mass, strength and power, significantly reduces walking speed [1]. The term “sarcopenia” was first used by Irwin H. Rosenberg [2] in the late 1980s to describe age-related loss of skeletal muscle mass. In literal translation, this term means “lack of body/meat”. Muscle mass decreases on average by 1–2% per year after about 50 years of age with an acceleration in the following years [3]. The muscle strength of a 70-year-old can be reduced by 20–40% compared to a 20-year-old and up to 50% less in a 90-year-old [4]. Changes in muscle tissue are both qualitative and quantitative. This means that not only does muscle mass and cross-sectional area decrease but muscle quality is also affected, whose indicator is the so-called specific strength (strength per unit of muscle cross-sectional area or unit of muscle mass) [5]. 

The mechanism of sarcopenia and possibilities of using biological markers as tools for diagnosis, treatment and prevention of sarcopenia are still being discussed. In the last review on this topic, 119 different biomarkers were identified, but whether clear associations between biomarkers and muscle function exist still requires future work [6]. Sarcopenia associated with processes that are direct consequences of aging is called primary sarcopenia, while that occurring in combination with diseases leading to impaired function and decrease in muscle mass (e.g., stroke, cancer) is secondary sarcopenia [7]. Primary sarcopenia and muscle atrophy due to immobilization (e.g., due to illness) are of a slightly different nature. Sarcopenia proceeds gradually; it mainly concerns type II fast-twitch fibers (the disappearance of type I fibers also accelerates after 80 years of age), is associated with a decrease in the number of muscle fibers and atrophy of individual fibers and is often accompanied by impaired motor neuron function [8]. These types of changes affect the ability to perform dynamic activities that require the generation of muscle power, including slowing down the gait [9]. In the case of atrophy due to immobilization, this process is more violent, it affects mainly slow-twitch fibers (type I) and is associated with a reduction in the cross section of the fibers rather than a reduction in their number. The function of motor neurons is most often preserved [10].

In 2010, the European Working Group of Sarcopenia in Older People (EWGSOP) expanded the definition to recognize sarcopenia as a syndrome characterized by progressive and generalized loss of skeletal muscle mass and strength with the risk of adverse consequences such as physical disability, decreased quality of life and death. At the same time, EWGSOP experts suggested using criteria such as low muscle mass, reduction of muscle strength and functional performance to diagnose sarcopenia [11]. In 2019, the EWGSOP updated the operational diagnostic criteria for identification of people with sarcopenia (EWGSOP2) [12]. The main focus was put on small muscle strength as a key characterization of sarcopenia. The other criteria are low muscle mass or poor quality and limited functional performance. Among the two proposed low performance measures of sarcopenia are gait speed and the Timed Up and Go (TUG) test. Both are measured in seconds and may also be used for the assessment of muscle function, stability, frailty and prediction of incident disability [13,14,15,16,17]. For example, the gait speed cut-off point of 1 m/s is used in the definition of sarcopenia by the International Working Group on Sarcopenia [18].

Gait speed is the most important component in assessing function in older adults. It is one of the criteria for frailty syndrome [19] and sarcopenia [12]. Gait speed is also an essential part of a battery of tests used to evaluate functions in older age (e.g., SPPB, Senior Fitness Test). 

The heterogeneity of the methods of gait speed assessment (e.g., different length of the walking path, different gait path—straight versus with turn, as in TUG) means that data from various studies collected in different databases are not always possible to use in joint scientific studies, especially in meta-analyses. A method that allows the results of various gait speed evaluation tests to be converted may also be useful in clinical practice. It may allow tracking the progression of the decline in function even if different tests were used for a given patient over a period of time. We decided to create a way to convert 4 m walking speed and the TUG test because these are the two most commonly used tests to assess gait speed in both clinical practice and research [20].

Usage of any of those tools requires recalculation for the second one for subsequent comparisons, e.g., in the meta-analyses. The simple linear regression has been used for such comparisons in several previous studies, but the appropriateness of such an approach has not been verified [21,22,23]. We hypothesised that the relationship between the two tests may not be linear. Therefore, the aim of the present study is to find the most appropriate model describing the relationship between these two popular measures of physical function. 

## 2. Material and Methods

### 2.1. Subjects

The study was performed in 450 (315 women and 135 men) consecutive outpatients of the Geriatric Clinic of the Medical University of Lodz, Poland, aged 70 to 92 years who volunteered to participate in the study. Predominance of women in the studied group results from the demographic profile of the Polish population. Medical review was performed with each subject. All participants underwent routine physical and mental examinations. Patients were measured and weighed on RADWAG personal weight scales (WPT60 150OW) (Radwag Balances and Scales, Radom, Poland). Body Mass Index (BMI) was calculated by dividing body weight by height in meters squared. The inclusion criteria were age 70 years and over, logical contact that allows understanding the instructions, ability to walk, and written consent to participate in the study. We excluded patients who were not able or refused to perform necessary tests, not able to stand or with serious psycho-cognitive impairment. 

The study was approved by the Bioethics Committee of the Medical University of Łódź (RNN/647/14/KB) and complies with the Declaration of Helsinki and Good Clinical Practice Guidelines. Written informed consent was obtained from all individual participants included in the study.

### 2.2. Physical Performance

To assess physical function, the TUG test and gait speed were used. In the TUG test, the time that a participant needs to rise from a chair, walk 3 m, turn around, walk back and sit down on the chair was recorded [23]. During the test, the person should wear her/his regular comfortable footwear and use any mobility aids (e.g., cane or frame) that they would normally require. Ten seconds or less indicate normal mobility, 11–20 s are within normal limits for frail older patients, and greater than 20 s means disability and need of assistance in transferring. A score of 14 s or more indicates an increased risk of falls. Research has shown the Timed up and Go test has good validity and interrater and intrarater reliability [22,24].

Gait speed was assessed with the 4 m Walking Test [13]. Patients were asked to walk at their usual speed on a 4 m flat, unobstructed track marked out with a tape. A stopwatch was used to record the time taken to complete the course. Patients were allowed to use their normal walking aids if required. The test demonstrated very good test–retest and interobserver reliability [25].

### 2.3. Statistical Methods

Statistical analysis was performed using the Statgraphics Centurion 18 version 18.1.13 software (Statgraphic Technologies, Inc., The Plains, Virginia, USA). Data were verified for normality of distribution. The results of gait speed complied with the normal distribution, while the distribution of TUG results was not. The one-way analysis of variance (ANOVA) and Mann–Whitney test were used to check the differences between men and women. To calculate the correlation between TUG and gait speed, different alternative models were compared to obtain the highest R-squared values. Numeric data are presented as mean ± SD (standard deviation) and as median and quartiles. Statistical significance of results was accepted at *p* < 0.05.

## 3. Results

A total of 450 outpatients of the Geriatric Clinic who performed both TUG and gait speed were included in the study. Mean age was 79.3 ± 5.4 years; 315 participants (70%) were women. 

The median education was 12 years; mean Body Mass Index was 27.2; 16.8% were active smokers. Among 450 participants, 68.6% were diagnosed with arterial hypertension, 59.5% had hypercholesterolemia, 19.2% suffered from type 2 diabetes mellitus, 10.6% of individuals had myocardial infarction, 10.4% had stroke in the past, 9.5% had history of past or present cancer. The group of 9.4% of the subjects reported chronic obstructive pulmonary disease, 45% had osteoarthritis, 21.2% osteoporosis, depression was diagnosed in 18.3%, while chronic congestive heart failure in 36.2% of participants. The proportion of 23.6% of the subjects had taken beta-blockers, 16.6% calcium channel blockers, 22.5% diuretics and 25.4% took other antihypertensive agents. The proportion of 23.4% of the subjects took statins, and 18.4% used anti-diabetic drugs.

Mean TUG test result was 8.60 ± 3.64 s, mean gait speed was 1.34 ± 0.41 m/s. Age, Timed Up and Go and gait speed in studied women and men are shown in Table 1. 

To find the most appropriate way to convert TUG and gait speed, the TUG test of 450 subjects was correlated with their gait speed. Different alternative models were compared to obtain the highest R-squared values. The Reciprocal-Y model (R-squared = 71.9%)(Figure 1a) showed the highest performance, followed by the Logarithmic-Y square root-X model (R-squared = 69.3%) and the Reciprocal-Y square root-X model (R-squared = 69.1%). R-squared for linear model was 49.5%. 

For the selected reciprocal model, the correlation coefficient was 0.85, and the equation of the fitted model was:Gait speed (m/s) = 1/(−0.0160767 + 0.101386 × TUG)

For the obtained equation: standard error (SE) for intercept = 0.0280; SE for slope = 0.00299; standard error of estimation (SEE) = 0.231; significance of the model—*p* < 0.001. There were no correlations between the residuals versus the values of X, the residuals versus the predicted values of Y and the residuals versus observation number.

Alternatively, the Reciprocal-X model may be used:TUG (s) = 2.53541 + 7.08702/Gait speed

For the selected reciprocal model, 0.8 m/s gait speed corresponded with 11.4 s in the TUG test while gait speed of 1 m/s with TUG of 9.6 s. Fourteen seconds in TUG corresponded with 0.71 m/s of gait speed, while 20 s in TUG corresponded with a gait speed of 0.5 m/s. 

The relationship between the two mobility variables was essentially identical when analysed separately by sex. The Reciprocal-Y model (R-squared = 61.5%) (Figure 1b) was slightly worse compared to the logarithmic-Y square root-X model (R-squared = 61.8%) and was followed by the multiplicative model (R-squared = 61.1%) in women. R-squared for the linear model was 47.0%. The equation of the reciprocal model for women was:Gait speed (m/s) = 1/(−0.0404068 + 0.104866 × TUG
(SE for intercept = 0.0418; SE for slope = 0.00469; SEE = 0.235; significance of the model—*p* < 0.001).

In men, the Reciprocal-Y model (R-squared = 83.5%) (Figure 1c) showed the highest performance, followed by the Reciprocal-Y squared-X model (R-squared = 82.1%) and the Reciprocal-Y square root-X model (R-squared = 80.9%). R-squared for the linear model was 58.0%. The equation of the reciprocal model for men was:Gait speed (m/s) = 1/(−0.00722707 + 0.0990616 × TUG)
(SE for intercept = 0.0392; SE for slope = 0.00382; SEE = 0.224; significance of the model—*p* < 0.001).

In 368 subjects with TUG ≤ 10 s, the Reciprocal-Y model (R-squared = 34.4%) (Figure 2a) was slightly worse compared to the logarithmic-Y squared-X model (R-squared = 35.8%). R-squared for the linear model was 34.2%. The equation of the reciprocal model for women was:Gait speed (m/s) = 1/(0.1389 + 0.0798216 × TUG
(SE for intercept = 0.0428; SE for slope = 0.00577; SEE = 0.136; significance of the model—*p* < 0.001).

In 82 subjects with TUG > 10 s, the Reciprocal-Y model (R-squared = 57.0%) (Figure 2b) was slightly worse compared to the logarithmic-Y square root-X model (R-squared = 58.4%). R-squared for the linear model was 48.8%. The equation of the reciprocal model for women was:Gait speed (m/s) = 1/(−0.0319556 + 0.103423 × TUG)
(SE for intercept = 0.153; SE for slope = 0.01005; SEE = 0.460; significance of the model—*p* < 0.001).

## 4. Discussion

Sarcopenia is one of the most important health problems at advanced age [1]. Sarcopenia affects the deterioration of functional performance, is a significant component of the frailty syndrome, is associated with an increased risk of institutionalization (placement in a nursing home), has a worse prognosis in the course of many diseases (including COVID-19) and entails the risk of death [10]. Overlapping with malnutrition and frailty, it is the leading cause of declining physical functioning [18]. The most important manifestations of sarcopenia are disability and recurrent falls [26]. Increasing life expectancy and the percentage of older individuals throughout the world make the problem of sarcopenia an important public health issue [27]. 

In this study, we have assessed the relationship of TUG with gait speed. Both measures are widely and interchangeably used in geriatric research and in sarcopenia definitions [12,13,28,29]. The relationship between the two measures has been considered as linear when recalculating the data [22,23]. Present data clearly indicate the nonlinear nature of the relationship between these variables.

Several evaluation tools have been proposed to assess gait, balance, and transfer in older adults [30]. The ability to walk is the basis for many activities of daily living and is necessary for independence. Walking speed is an easily accessible screening tool to assess functional capacity. There are many protocols to assess walking speed which vary in terms of path length (2–40 m), start (static versus dynamic), path (straight versus with turn, e.g., the TUG test), speed (self-selected versus maximal) and instruction (e.g., “walk at a comfortable pace” versus “walk as if you are taking a stroll through the park”) [20]. The assessment at a usual pace over 4 m has been recommended as a quick, safe, inexpensive and highly reliable instrument to predict adverse outcomes in community-dwelling older people [13].

At the same time, a commonly used screening tool in clinical practice and one of the most popular tools used to assess functional abilities, including gait assessment in the elderly, is the TUG test [31]. TUG, and its several modifications shows a good reliability and concurrent validity with other performance measures [24]. Therefore, the creation of a tool for converting test results (walking speed vs. the TUG test) makes it possible to use the results of studies assessing these two performance indicators, e.g., in meta-analyzes or in retrospective analyses comparing the results of studies conducted with the use of either of these tools.

Both TUG and gait speed are widely used for the assessment of physical functioning, sarcopenia and frailty in different medical conditions and in older adults. Their predictive value for incident disability seems comparable [14,15]. Odds ratios in predicting multiple geriatric outcomes were equivalent for gait speed and TUG in 457 older adults [14]. In the Irish Longitudinal Study on Aging (*n* = 1664), TUG and usual gait speed had similar predictive ability in relation to incident disability in basic ADL and IADL after 2 years [15]. Both TUG and gait speed were reliable outcome measures for use with people with Alzheimer’s disease [32]. Bridenbaugh et al. suggested that the discrepancy between real and imagined TUG performances may be a surrogate marker of disturbed higher-level gait control in older adults [33].

In the older Chinese community, slower gait speed and a poorer TUG test were similarly associated with osteosarcopenic obesity in men and women. However, compared to grip strength and gait speed, only the TUG test had significant differences between osteosarcopenic obesity and obesity, osteosarcopenic obesity and osteopenic obesity, osteosarcopenic obesity and sarcopenic obesity in women [34]. Nevertheless, some results suggest that gait speed may be more representative of the whole motor ability of frail older patients than the TUG [35]. The TUG may be susceptible to the floor effect in subjects with good physical functioning, while gait speed may be useful across the full spectrum of functionality in older adults [36]. In the systematic review, Barry and colleagues concluded that the TUG test has limited ability to predict falls in community-dwelling older adults [37]. Procedural variations, e.g., seat height, can significantly impact the reliability of the TUG test [38]. Similarly, sarcopenia prevalence varies with muscle strength definitions with population-specific performing better than standard cutoffs to obtain estimates of sarcopenia prevalence using the EWGSOP2 clinical algorithm [39]. Furthermore, in the Canadian Study of Health and Aging, physical performance measures proved infeasible in many subjects (29.3% for the Timed Up and Go [TUG], 35.9% for the functional reach [FR]). Cognitive impairment was the most important determinant of inability to complete the tests. Results support the observation that subsequent studies of measurement instruments typically reveal lower performance than the original reports [40]. Likewise, in the Canadian Longitudinal Study on Aging (CLSA), the relative reliability for grip strength was excellent (intraclass correlation coefficient — ICC = 0.95); the TUG and single-leg stance tests had good reliability (ICC = 0.80 or 0.78–0.82, respectively); gait speed and the chair-rise test had moderate reliability (ICC = 0.64 for both) [41].

Previous studies showed mean and median values of TUG and gait speed or used Pearson and Spearman correlations between the two measures [21,22,23,42]. Herman et al. found normal distribution of TUG results in 265 healthy older adults. The authors concluded that TUG does not suffer from the ceiling effect limitations and is apparently related to executive function [22]. In a study in 79 orthopaedic rehabilitation inpatients, the relationship between gait time and TUG was linear both at admission and discharge [43]. The Pearson correlation coefficient between the TUG test and the 10 m walk time was r = 0.67 (*p* < 0.01) in 100 Persian community-dwelling older adults [21]. In the classic study of Podsiadlo and Richardson, the correlation of TUG and gait speed was r = −0.61 [23]. Pearson’s correlation coefficient between the usual gait speed (UGS) and the TUG test was r = −0.66 (*p* < 0.001) in elderly vestibular patients [44]. Functional gait assessment (FGA) correlated with TUG (*r*= −0.84, *p* < 0.001)(Spearman correlation coefficient) in 35 community-dwelling older adults [45].

In our study, the results of gait speed complied with the normal distribution, while the distribution of TUG results was clearly not normal with few results spread for TUG results of 15 s and longer. Not surprisingly, the relationship between the two values was nonlinear, with the reciprocal model (R-squared = 71.9%) showing the highest performance to predict gait speed from TUG, much higher than the linear one. Similarly, a nonlinear relationship between gait speed and falls was found by Quach et al., with a greater risk of outdoor falls in fast walkers and a greater risk of indoor falls in slow walkers compared to those with normal gait speeds (1.0–1.3 m/s) [46].

Data from the obtained model seem to be in accord with studies showing mean (median) values of both measures: (1) 0.88 m/s gait speed corresponded with 12.3 s in the study of Viccaro et al., in older adults with a mean age of 74 years [14]; (2) 1.3 m/s gait speed corresponded with 9.5 s in the study of Herman et al. in healthy older adults with a mean age of 76.4 years [22]; (3) 0.8 m/s gait speed corresponded with 10.9 s in the study of Kramer et al., in patients with cardiac implantable devices [47]; (4) 0.8 m/s gait speed corresponded with 12.6 s in the TUG test with a gait speed of 1 m/s with TUG of 9.9 s. in 163 older subjects [48]; (5) 0.78 m/s gait speed corresponded with 14.1 s of TUG and 1.1 m/s gait speed corresponded with 9.7 s of TUG in older Chinese adults [34]; (6) 0.52 m/s gait speed corresponded with 28 s of TUG and 0.66 m/s gait speed corresponded with 20 s of TUG in patients with Alzheimer’s disease [32]; (7) 0.8 m/s gait speed corresponded with 11.2 s in the TUG test while gait speed of 0.97 m/s with TUG of 10.1 s in patients with Parkinson’s disease [49]; (8) 1.4 m/s gait speed corresponded with TUG of 5.3 s in Japanese community-dwelling older adults [50]. Therefore, the proposed model may be useful for calculations of gait speed or TUG in future studies, especially in meta-analyses.

Our study has some limitations. We were not able to conduct tests amongst people with severe dementia or those who due to mobility problems were not able to present at our clinic. The study was conducted in a Central European region, and results may be different in other populations and in different groups of patients. As the first report undertaking this issue, the study needs validation of the proposed equations, e.g., in an external sample. This problem should be considered in future research in this area. On the other hand, conformity with TUG–gait speed comparisons from previous studies suggests that the observed nonlinear relationship may be valid in different settings.

## 5. Conclusions

The relationship between the gait speed and TUG results is nonlinear with the reciprocal model (R-squared = 71.9%) showing the highest performance, much higher than the linear one (R-squared = 49.5%). For the selected reciprocal model, the correlation coefficient was 0.85, and the equation of the fitted model was:Gait speed = 1/(−0.0160767 + 0.101386 × TUG)

Therefore, the proposed model may be useful for recalculations of gait speed or TUG in future studies.

## Figures and Tables

**Figure 1 ijerph-19-12145-f001:**
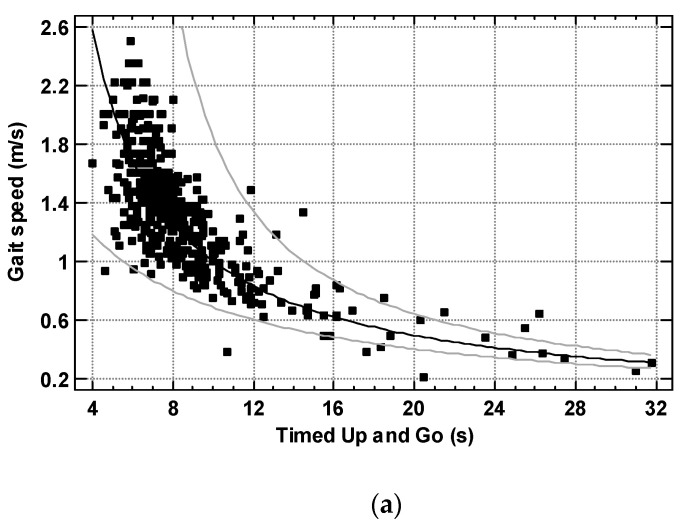
(**a**) the relationship of Timed Up and Go to gait speed in 450 subjects (regression line, 95% prediction limits); (**b**) the relationship of Timed Up and Go to gait speed in 315 women (regression line, 95% prediction limits); (**c**). the relationship of Timed Up and Go to gait speed in 135 men (regression line, 95% prediction limits).

**Figure 2 ijerph-19-12145-f002:**
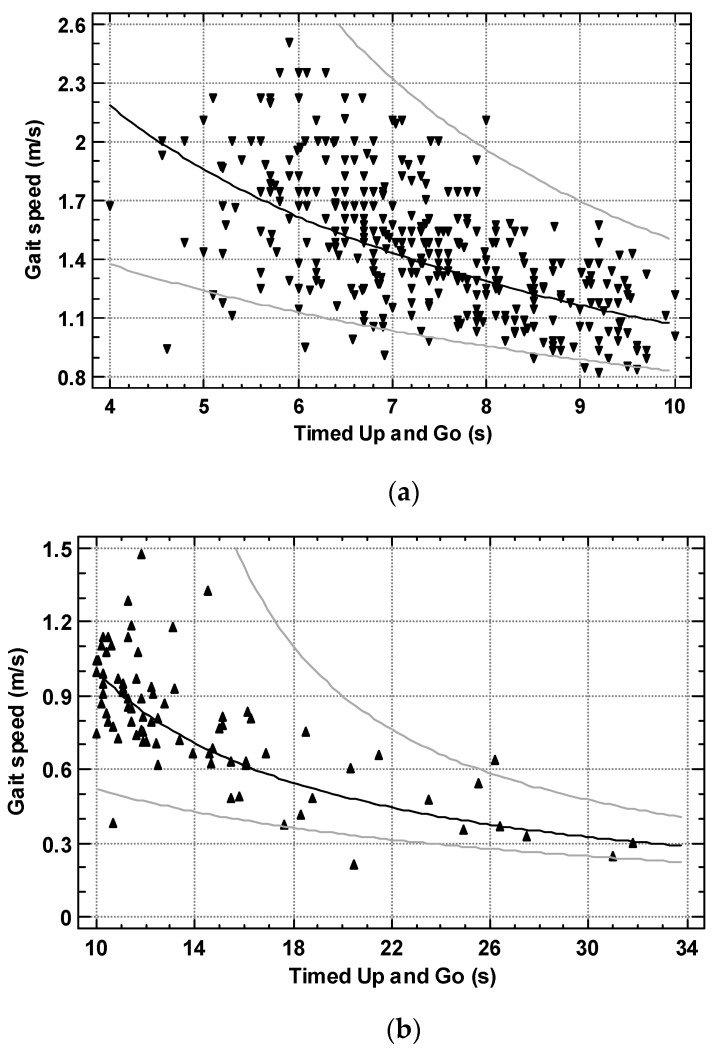
(**a**) the relationship of Timed Up and Go to gait speed in 368 subjects with TUG ≤ 10 s (regression line, 95% prediction limits); (**b**) the relationship of Timed Up and Go to gait speed in 82 subjects with TUG > 10 s (regression line, 95% prediction limits).

**Table 1 ijerph-19-12145-t001:** Age, Timed Up and Go and gait speed in studied women and men.

	All (*n* = 450)	Women (*n* = 315)	Men (*n* = 135)	Significance
Age (years)	79.3 ± 5.4 80 (75–83)	79.2 ± 5.3 79 (75–83)	79.6 ± 5.5 80 (75–84)	*p* = 0.56
TUG (seconds)	8.60 ± 3.64 7.6 (6.7–9.2)	8.46 ± 2.83 7.8 (6.8–9.3)	8.94 ± 5.05 7.4 (6.3–9.1)	*p* = 0.20
Gait speed (m/s)	1.34 ± 0.41 1.33 (1.07–1.6)	1.32 ± 0.38 1.3 (1.05–1.54)	1.38 ± 0.47 1.42 (1.11–1.67)	*p* = 0.13

The data are presented as mean ± SD and median (Q1–Q3).

## Data Availability

The datasets analyzed during the current study are available from the corresponding author on reasonable request.

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
