# Peer review of "A Proposed Method of Converting Gait Speed and TUG Test in Older Subjects"

_ijerph, 2022, doi:10.3390/ijerph191912145_

Round 1
Reviewer 1 Report
The aim of this study seems interesting, but the authors did not sufficiently justify its necessity and validity of these studies. I also have doubts about the practical application of the proposed equations, because their accuracy has not been validated, e.g. in an external sample. However, the results obtained in a relatively large group of people over the age of 70 enrich the current literature and may be of importance for future research on various gait speed tests and the possibility of comparing or converting their results. Before possible publication, the manuscript requires major additions and corrections.
Abstract: There is no clearly defined goal of the work.
The majority of the introduction concerns sarcopenia, which was not assessed in the participants of this study, while relatively little information is available about the GS and TUG tests, the relationship of which is the main topic of the work. The authors should describe both tests here, indicate their importance in identifying different clinical conditions, and clearly justify the need to convert the results of one test to the results of the other.
Material and methods:
Line 82, please provide the number of men and the number of women.
Line 115, the p-value is not the significance level. It should be "α = 0.05" instead of "p <0.05" or you can change this sentence to "Statistical significance of results was accepted at p <0.05".
Was the normality of the distribution of variables checked?
What test was used to check the differences between men and women (Table 1)?
Results:
Lines 120-129, for these results (except BMI) there is no description of the methods of obtaining them in the Material and methods section (medical interview, questionnaire?)
Table 1, please provide information (in the header or under the table) that the reported values ​​are mean ± standard deviation and median (first quartile - third quartile). With GS it should be m/s (Table 1 and line 130). It is advisable to enter a p-value instead of "NS".
For the obtained equation, the results of regression should be given, such as the significance of the model and the significance of its coefficients, standard error of coefficients (SE), standard error of estimation (SEE). An additional table with the results for all equations can be considered.
Have the assumption of normality and the absence of autocorrelation of the regression residuals been checked?
I suggest checking the significance of the differences between the observed and predicted values.
Figure 1, the lines in the graph are not described.
The discussion needs to be redrafted, most of the information is more suited to the introduction (lines 167-221). In this section, the authors should highlight the discussion of the results obtained for the GS and TUG tests, and evaluate them in relation to previous studies in similar populations (similarities / differences).
Lines 235-239, the non-linearity of the relationship between GS (m / s) and TUG (s) is not due to the non-normality of the distribution, in this case it is rather because walking speed is the ratio of path to time.
Lines 243-256, this information does not indicate to the reader "similarities" to the data obtained from the regression model. In the case of the research results of other authors, the values ​​estimated by the derived formulas can be given in parentheses. From my calculations, for GS = 0.88 the predicted TUG value is 10.59, the difference of almost 2 seconds from the observed value (TUG = 12.3) can be significant (p <0.05 even assuming n = 50 and SD = 5). Please check compliance with the references given.
Line 261, "cultures" replace with "populations".
The conclusions should be edited, because at this stage of the research the usefulness of the proposed model may raise doubts. Its accuracy has not been verified, and the differences between the observed values ​​and those predicted by the regression model have not been checked.
Author Response
Dear Editor,
Thank you for the review and the invitation to revise our manuscript entitled: “A proposed method of converting gait speed and TUG test in older subjects”.
We have revised the manuscript according to the Reviewers’ comments. We have tried to response to the Reviewers and information about changes are included below and marked in red in the text.
Dear Reviewers,
Thank you for all the constructive comments.
|
Reviewer 1 |
|
|
Comment |
Answer |
|
The aim of this study seems interesting, but the authors did not sufficiently justify its necessity and validity of these studies.
|
We have added explanation at the end of the introduction: Gait speed is the most important component in assessing function in older adults. It is one of the criteria for frailty syndrome [Fried] and sarcopenia [Cruz-Jentoft 2019]. Gait speed is also an essential part of a battery of tests used to evaluate functions in older age (e.g. SPPB, Senior Fitness Test). The heterogeneity of the method of gait speed assessment (e.g. different length of the walking path, different gait path - straight versus with turn, as in TUG) means that the use of data from various studies, collected in different databases, is not always possible to use in joint scientific studies, especially in meta-analyses. A method that allows the results of various gait speed evaluation tests to be converted may also be useful in clinical practice. It may allow to track the progression of the decline in function even if different tests were used for a given patient over a period of time. We decided to create a way to convert 4m walking speed and the TUG test because these are the two most commonly used tests to assess gait speed in both clinical practice and research [Middleton 2015]. and: We hypothesised that the relationship between the two tests may not be linear. |
|
I also have doubts about the practical application of the proposed equations, because their accuracy has not been validated, e.g. in an external sample |
Of course, as the first report undertaking this issue it should be confirmed in future studies. We have included this problem in the limitations of the study: As the first report undertaking this issue, the study needs validation of the proposed equations, e.g. in an external sample. This problem should be considered in future research in this area. |
|
Abstract |
|
|
There is no clearly defined goal of the work. |
We have added the goal in the abstract section: The aim of this study is to find the most appropriate model describing the relationship between these two popular measures of physical function. |
|
Introduction |
|
|
The majority of the introduction concerns sarcopenia, which was not assessed in the participants of this study, while relatively little information is available about the GS and TUG tests, the relationship of which is the main topic of the work. The authors should describe both tests here, indicate their importance in identifying different clinical conditions, and clearly justify the need to convert the results of one test to the results of the other. |
We have added a brief explanation on relationship of GS and sarcopenia and described the importance of the GS and TUG tests as suggested: Walking is a dynamic activity that requires the use of an appropriate muscle function. Age - related sarcopenia, associated with a gradual decrease in muscle mass, strength and power, significantly reduces walking speed [Lauretani 2003].
|
|
Material and methods: |
|
|
Line 82, please provide the number of men and the number of women. |
We have added this information: (315 women and 135 men) |
|
Line 115, the p-value is not the significance level. It should be "α = 0.05" instead of "p <0.05" or you can change this sentence to "Statistical significance of results was accepted at p <0.05". |
“Statistical significance of results was accepted at p <0.05.” has been corrected. |
|
Was the normality of the distribution of variables checked? |
The normality of the distribution of variables was checked and described in the Statistical methods and discussion: Data were verified for normality of distribution. In our study, the results of gait speed complied with the normal distribution while the distribution of TUG results was clearly not normal with few results spread for TUG results of 15 seconds and longer. |
|
What test was used to check the differences between men and women (Table 1)? |
“The one-way analysis of variance (ANOVA) and Mann-Whitney test were used to check the differences between men and women.” This additional information has been provided in the Statistical methods. |
|
Results |
|
|
Lines 120-129, for these results (except BMI) there is no description of the methods of obtaining them in the Material and methods section (medical interview, questionnaire?) |
Medical review was performed with each subjects. All participants underwent routine physical and mental examinations. |
|
Table 1, please provide information (in the header or under the table) that the reported values are mean ± standard deviation and median (first quartile - third quartile). With GS it should be m/s (Table 1 and line 130). It is advisable to enter a p-value instead of "NS". |
We have added this information to Table 1: mean ± SD median (Q1-Q3)
“GS in m/s” has been corrected. Exact p-values have been provided as suggested. |
|
For the obtained equation, the results of regression should be given, such as the significance of the model and the significance of its coefficients, standard error of coefficients (SE), standard error of estimation (SEE). An additional table with the results for all equations can be considered. |
All the requested data have been provided to the obtained equations.
We constructed an additional table with the results for all equations but it detracted from the legibility of the data. Therefore, we decided to present these equations in the text. |
|
Have the assumption of normality and the absence of autocorrelation of the regression residuals been checked? I suggest checking the significance of the differences between the observed and predicted values. |
Yes, there was no correlation between the regression residuals and presented data. The significance of the differences between the observed and predicted values has been checked – there were no significant differences. This information has been provided: „There were no correlations between the residuals versus the values of X, the residuals versus the predicted values of Y, and the residuals versus observation number.” |
|
Figure 1, the lines in the graph are not described. |
The lines in the graphs have been described as suggested. |
|
Discussion |
|
|
The discussion needs to be redrafted, most of the information is more suited to the introduction (lines 167-221). In this section, the authors should highlight the discussion of the results obtained for the GS and TUG tests, and evaluate them in relation to previous studies in similar populations (similarities / differences). |
We have moved some information form the discussion to the introduction as suggested. Nevertheless, we believe that to discuss similarities and differences of our data with previous studies those reports should be presented in the discussion.
|
|
Lines 235-239, the non-linearity of the relationship between GS (m / s) and TUG (s) is not due to the non-normality of the distribution, in this case it is rather because walking speed is the ratio of path to time. |
The results of gait speed complied with the normal distribution while the distribution of TUG results was clearly not normal with few results spread for TUG results of 15 seconds and longer. This information has been provided. |
|
Lines 243-256, this information does not indicate to the reader "similarities" to the data obtained from the regression model. In the case of the research results of other authors, the values estimated by the derived formulas can be given in parentheses. From my calculations, for GS = 0.88 the predicted TUG value is 10.59, the difference of almost 2 seconds from the observed value (TUG = 12.3) can be significant (p <0.05 even assuming n = 50 and SD = 5). Please check compliance with the references given. |
Of course, the data are not the same in different studies. From previous studies one may cite only the relationship between the two variables at given point and cited associations are correct. If we compare present data at a given point with those reported in previous studies, our data seem to be well inside reported values. For example, in our study, for the selected reciprocal model, 0.8 m/s gait speed corresponded with 11.4s in TUG test. In others reporting that speed, 0.8 m/s gait speed corresponded with 10.9s in the study of Kramer et al., 0.8 m/s gait speed corresponded with 12.6s in TUG test in 163 older subjects (Sosowska et al.), 0.8 m/s gait speed corresponded with 11.2s in TUG test in patients with Parkinson’s disease (Li et al.). Therefore, we believe that reported equations reflect general association between variables, with potential differences between given studies. This potential and probably unavoidable bias has been acknowledged in the limitations of the study: “The study was conducted in a Central-European region and results may be different in other populations as well as in different groups of patients.” |
|
Line 261, "cultures" replace with "populations". |
Has been modified as suggested. |
|
Conclusions |
|
|
The conclusions should be edited, because at this stage of the research the usefulness of the proposed model may raise doubts. Its accuracy has not been verified, and the differences between the observed values and those predicted by the regression model have not been checked. |
We do agree that proposed model should be verified in external analyses and future studies.
|
|
Reviewer 2 |
|
|
Comment |
Answer |
|
Introduction |
|
|
Why the authors try to relate the results of the Timed Up and Go (TUG) test with the results of the gait speed test at 4 m ? Both of them represent varied physiological abilities and varied functional background. A paragraph justifying this rationale is needed. |
We have added explanation at the end of the introduction: Gait speed is the most important component in assessing function in older adults. It is one of the criteria for frailty syndrome [Fried] and sarcopenia [Cruz-Jentoft 2019]. Gait speed is also an essential part of a battery of tests used to evaluate functions in older age (e.g. SPPB, Senior Fitness Test). The heterogeneity of the methods of gait speed assessment (e.g. different length of the walking path, different gait path - straight versus with turn, as in TUG) means that the use of data from various studies, collected in different databases, is not always possible to use in joint scientific studies, especially in meta-analyses. A method that allows the results of various gait speed evaluation tests to be converted may also be useful in clinical practice. It may allow to track the progression of the decline in function even if different tests were used for a given patient over a period of time. We decided to create a way to convert 4m walking speed and the TUG test because these are the two most commonly used tests to assess gait speed in both clinical practice and research [Middleton 2015]. and: We hypothesised that the relationship between the two tests may not be linear.
|
|
Regarding the mechanism od sarcopenia, I would suggest to visit two new relevant review papers concerning this issue: - Jones, R.L.; Paul, L.; Steultjens, M.P.M.; Smith, S.L. Biomarkers associated with lower limb muscle function in individuals with sarcopenia: a systematic review. J. Cachexia. Sarcopenia Muscle 2022, doi:10.1002/jcsm.13064. - Bilski, J.; Pierzchalski, P.; Szczepanik, M.; Bonior, J.; Zoladz, J.A. Multifactorial mechanism of sarcopenia and sarcopenic obesity. Role of physical exercise, microbiota and myokines. Cells 2022, 11, doi:10.3390/cells11010160. |
According to suggestion we have added these two references: The mechanism of sarcopenia and possibilities of using biological markers as tools for diagnosis, treatment and prevention of sarcopenia are still being discussed. In the last review on this topic, 119 different biomarkers were identified, but whether clear associations between biomarkers and muscle function exist still requires future work [Jones 2022} and: These types of changes affect the ability to perform dynamic activities that require the generation of muscle power, including slowing down the gait [Bilski 2022].
|
|
The Introduction section is missing a paragraph concerning the impact of sarcopenia on the power generating capabilities of ageing people. During dynamic physical activities muscle power is more important than the maximal muscle force, measured in isometric contractions (see also the above mentioned review papers). |
We have supplemented the introduction with the recommended information: Walking is a dynamic activity that requires the use of an appropriate muscle function. Age - related sarcopenia, associated with a gradual decrease in muscle mass, strength and power, significantly reduces walking speed [Lauretani 2003]. And: Sarcopenia proceeds gradually, it mainly concerns type II fast-twitch fibers (the disappearance of type I fibers also accelerates after 80 years of age), is associated with a decrease in the number of muscle fibers and atrophy of individual fibers and is often accompanied by impaired motor neuron function {Frontera, 2012 #242}. These types of changes affect the ability to perform dynamic activities that require the generation of muscle power, including slowing down the gait [Bilski 2022]. |
|
Results |
|
|
The Figure 1. Please present a figure 1 B and 1 C presenting the same calculations but for the group of men and women separately (Figure 1 B and 1 C, respectively). |
Figures 1B and 1C have been added as suggested. |
|
An additional Figure 2, would be useful to add to show the same calculations but for the sub-group of patients: (1) with TUG up to 10 s (“physiological range); (2) with the group of patients + 10 s up to 20 s and (3) for the group with TUG above 20 s (see page 3, lines 98-104). |
As only 11 subjects had TUG > 20 sec., we have added Figure 2A with TUG up to 10 sec. and Figure 2B with TUG above 10 sec. |
|
When presenting the equation for the gait speed (see, e.g., page 1, line 22 and the other parts of the manuscript) please show its units as well (m x s-1) ? Are they in accordance with the units of the variables in this equation? |
The units have been presented as suggested. |
|
The Results section: the Table 1 needs more explanations. For example the data 79.4 +/- 5.4 and 80 (75-83). Is it a mean +/- SD and the range of results? Rather not, since the range of the studied people was 70-92 y (see page 1 line 16). This issue needs to be clarified. |
We have added this information to Table 1: mean ± SD median (Q1-Q3) |
|
Discussion |
|
|
A paragraph is needed to clarify the issue of the functional dependence / independence of the results of the Timed Up and Go (TUG) test and the results of the gait speed test at 4 m. Why replace the one by the other? |
Suggested information has been added both in the introduction and discussion. |

Reviewer 2 Report
Ms.: ijerph-1902543
The authors studied the relationship between the results of the Timed Up and Go (TUG) test with the results of the gait speed test at 4 m in a group of elderly people (70-92 years old).
As the main outcome of this study, the authors at the Figure 1 presented the relationship between the TUG results and the gait speed test for 450 subjects.
The authors studied the relationship between the results of the Timed Up and Go (TUG) test the results of the gait speed test at 4 m using varied modeling approaches. The highest performance was obtained using the reciprocal-Y (R-squared = 83.5%), whereas the R-squared for linear model was 58.0 %. Therefore the authors concluded that: That the relationship between the results of the Timed Up and Go (TUG) test the results of the gait speed test at 4 m obtained in the studied elderly population is not linear and the proposed reciprocal model may be useful for recalculations of gait speed or TUG in future studies.
In genal this paper contains some interesting results but in my opinion some additional data would increase its value. Therefore I have some comments to this manuscript in its present form:
1. The Introduction: Why the authors try to relate the results of the Timed Up and Go (TUG) test with the results of the gait speed test at 4 m ? Both of them represent varied physiological abilities and varied functional background. A paragraph justifying this rationale is needed.
2. The Introduction: Regarding the mechanism od sarcopenia, I would suggest to visit two new relevant review papers concerning this issue:
Jones, R.L.; Paul, L.; Steultjens, M.P.M.; Smith, S.L. Biomarkers associated with lower limb muscle function in individuals with sarcopenia: a systematic review. J. Cachexia. Sarcopenia Muscle 2022, doi:10.1002/jcsm.13064.
Bilski, J.; Pierzchalski, P.; Szczepanik, M.; Bonior, J.; Zoladz, J.A. Multifactorial mechanism of sarcopenia and sarcopenic obesity. Role of physical exercise, microbiota and myokines. Cells 2022, 11, doi:10.3390/cells11010160.
3. The Introduction section is missing a paragraph concerning the impact of sarcopenia non the power generating capabilities of ageing people. During dynamic physical activities muscle power is more important than the maximal muscle force, measured in isometric contractions (see also the above mentioned review papers).
4. The Results section. The Figure 1. Please present a figure 1 B and 1 C presenting the same calculations but for the group of men and women separately (Figure 1 B and 1 C, respectively).
5. The Results section: An additional Figure 2, would be useful to add to show the same calculations but for the sub-group of patients: (1) with TUG up to 10 s (“physiological range); (2) with the group of patients + 10 s up to 20 s and (3) for the group with TUG above 20 s (see page 3, lines 98-104).
6. When presenting the equation for the gait speed (see, e.g., page 1, line 22 and the other parts of the manuscript) please show its units as well (m x s-1) ? Are they in accordance with the units of the variables in this equation?
7. The Results section: the Table 1 needs more explanations. For example the data 79.4 +/- 5.4 and 80 (75-83). Is it a mean +/- SD and the range of results? Rather not, since the range of the studied people was 70-92 y (see page 1 line 16). This issue needs to be clarified.
8. Discussion section:
A paragraph is needed to clarify the issue of the functional dependence / independence of the results of the Timed Up and Go (TUG) test and the results of the gait speed test at 4 m. Why replace the one by the other?
Author Response

(The authors gave the same response as above.)

Round 2
Reviewer 1 Report
I have no further comments. Thank you for your comprehensive answers and satisfactory corrections. I hope the criticism was not too troublesome and helped to improve this manuscript.
Reviewer 2 Report
I have no further comments to this manuscript.